# How informative were early SARS-CoV-2 treatment and prevention trials? a longitudinal cohort analysis of trials registered on ClinicalTrials.gov

**Nora Hutchinson**[1], **Katarzyna Klas**[2], **Benjamin G. Carlisle**[3], **Jonathan Kimmelman**[1], **Marcin Waligora**[2]*

**1** Studies of Translation, Ethics, and Medicine (STREAM), Biomedical Ethics Unit, McGill University, Montreal, Québec, Canada, **2** Faculty of Health Sciences, Research Ethics in Medicine Study Group (REMEDY), Jagiellonian University Medical College, Krakow, Poland, **3** BIH QUEST Center for Transforming Biomedical Research, Berlin Institute of Health at Charité (BIH), Berlin, Germany

* m.waligora@uj.edu.pl

## Abstract

### Background

Early in the SARS-CoV-2 pandemic, commentators warned that some COVID trials were inadequately conceived, designed and reported. Here, we retrospectively assess the prevalence of informative COVID trials launched in the first 6 months of the pandemic.

### Methods

Based on prespecified eligibility criteria, we created a cohort of Phase 1/2, Phase 2, Phase 2/3 and Phase 3 SARS-CoV-2 treatment and prevention efficacy trials that were initiated from 2020-01-01 to 2020-06-30 using ClinicalTrials.gov registration records. We excluded trials evaluating behavioural interventions and natural products, which are not regulated by the U.S. Food and Drug Administration (FDA). We evaluated trials on 3 criteria of informativeness: potential redundancy (comparing trial phase, type, patient-participant characteristics, treatment regimen, comparator arms and primary outcome), trials design (according to the recommendations set-out in the May 2020 FDA guidance document on SARS-CoV-2 treatment and prevention trials) and feasibility of patient-participant recruitment (based on timeliness and success of recruitment).

### Results

We included all 500 eligible trials in our cohort, 58% of which were Phase 2 and 84.8% were directed towards the treatment of SARS-CoV-2. Close to one third of trials met all three criteria and were deemed informative (29.9% (95% Confidence Interval 23.7–36.9)). The proportion of potentially redundant trials in our cohort was 4.1%. Over half of the trials in our cohort (56.2%) did not meet our criteria for high quality trial design. The proportion of trials with infeasible patient-participant recruitment was 22.6%.

**Data Availability Statement:** The datasets generated and analysed during the current study

are available in the Open Science Framework repository, https://osf.io/fp726/.

**Funding:** This study was funded by the National Science Center, Poland, UMO 2020/01/0/HS1/00024 (www.ncn.gov.pl). Authors received the funding: MW, KK. The funder had no role in study design, data collection and analysis, decision to publish, or preparation of the manuscript.

**Competing interests:** Marcin Waligora reports personal fees from Advisory Bioethics Council, Sanofi outside the submitted work. This does not alter our adherence to PLOS ONE policies on sharing data and materials. Other authors have declared that no competing interests exist.

## Conclusions

Less than one third of COVID-19 trials registered on ClinicalTrials.gov during the first six months met all three criteria for informativeness. Shortcomings in trial design, recruitment feasibility and redundancy reflect longstanding weaknesses in the clinical research enterprise that were likely amplified by the exceptional circumstances of a pandemic.

## Introduction

Starting in early 2020, commentators warned of COVID-19 clinical trial design deficiencies and lack of coordination of research efforts [1–4]. The large volume of small trials investigating the efficacy of repurposed medications, such as hydroxychloroquine, in the treatment of COVID-19, drew particular attention [5,6]. Such studies confounded an effective public health response by producing spurious findings, or by diverting patients and resources from well designed and executed studies.

Appropriate design, implementation and reporting is captured by the concept of trial "informativeness" [3,7]. For a trial to be informative to clinical practice, it must fulfill five conditions [3,7]. First, it must ask a clinically important question. Second, it must be designed to provide a clear answer to that question. Third, it must have both a feasible enrollment target and primary completion timeline. Fourth, it must be analyzed in a manner that supports statistically valid inference. Fifth, it must report results in a complete and timely manner [3,7].

In the following longitudinal cohort analysis of SARS-CoV-2 treatment and prevention trials registered within the first 6 months of 2020, we assess three features of an informative clinical trial—potential redundancy, design quality and feasibility of patient-participant recruitment. Multiple cross-sectional analyses and systematic reviews of SARS-CoV-2 treatment and prevention trials have been performed [2,5,6,8–11], reporting on intervention types, study characteristics and choice of outcome measure. We go beyond a description of trial characteristics and provide the first in-depth evaluation of SARS-CoV-2 trial informativeness. Knowing the prevalence of potentially uninformative trials conducted in the early stages of the pandemic can help motivate the development of more effective research policy in anticipation of future public health crises.

## Methods

### Sample, design and trials selection

Our cohort consisted of interventional SARS-CoV-2 treatment and prevention trials registered on ClinicalTrials.gov with a start date between 2020-01-01 and 2020-06-30. We included "Completed", "Terminated", "Suspended", "Active, not recruiting", "Enrolling by invitation" and "Recruiting" Phase 1/2, Phase 2, Phase 2/3 and Phase 3 interventional clinical trials testing an efficacy hypothesis in their primary outcome. We included trials evaluating any of the following interventions: drug, biological, surgical, radiotherapy, procedural or device. We excluded trials evaluating behavioural interventions, trials of natural products and Phase 1 trials, all of which have no legal requirement to register on ClinicalTrials.gov [12]. See S1 File for complete inclusion/exclusion criteria. Trial inclusion and exclusion criteria were independently assessed by two researchers (KK & LZ), with disagreements resolved by an arbiter (NH or MW). We did not perform a sample size calculation, as we included all trials meeting our eligibility criteria within our designated sampling timeframe.

## Data curation

We downloaded clinical trial data directly as a zipped folder of XML files from the web front-end of ClinicalTrials.gov on 2020-12-01 and again on 2021-01-04 (see S2 File for Clinical-Trials.gov search criteria). This allowed us to evaluate data at the 6-month mark (from date of trial start) for all trials in our cohort (see S3 File for data directly downloaded from Clinical-Trials.gov). Additional items requiring human curation were independently assessed and coded by two researchers (KK & LZ), these included: i) treatment type (according to the World Health Organization (WHO) COVID-19 Classification of treatment types [13]); ii) illness severity (as stated by the study investigators or guided by the WHO disease severity classification [14]); iii) location of care (ambulatory, hospitalized, intensive care, unclear/not stated); iv) presence of a placebo or standard of care arm; and, v) type of primary outcome (clinical, surrogate, procedural) (see S4 File for additional double-coded data points). Disagreements were resolved by an arbiter (NH or MW) (Please see S1 Table for inter-rater agreement).

## Measures

Trials were assessed based on three elements of informativeness: i) potential redundancy (as a marker of trial importance); ii) trial design quality; and iii) successful patient-participant recruitment (as a marker of feasibility). Assessment criteria for each element were designed based on face validity and easy applicability over a large trial sample.

**Potential redundancy.** We assessed potential redundancy by evaluating non-redundancy of the trial hypothesis. Non-redundancy was defined as: absence of a trial of the same phase, type of trial (SARS-CoV-2 prevention versus treatment), patient-participant characteristics (including location of care, disease severity and age of trial participants), regimen (including interventions used in combination in a single arm), comparator arm(s) and primary outcome (evaluating primary outcome domain and specific measurement, based on framework from [15]) launched prior to the start date of the trial of interest (as indicated in the registration record active at the 6-month mark since trial start) (S5 File). Only the trial with the later start date was labelled as potentially redundant. The assessment was independently performed by two raters (NH & KK), with disagreements resolved by an arbiter (MW of BC). We performed an additional *post hoc* assessment applying a broad criterion for trial similarity, which we defined as presence of a trial with an earlier start date of the same type, phase, patient-partici-pant characteristics and treatment regimen.

**Design quality.** We analyzed trial design quality for those studies in our sample that were aimed at informing clinical practice–namely Phase 2/3 and Phase 3 trials. Based on the U.S. Food & Drug Association (FDA) May 2020 guidance document for SARS-CoV-2 drug and biological treatment and prevention trials [16], we considered a trial to be well-designed if it was randomized, placebo-controlled or with a standard of care comparator arm, double-blinded and included participants aged 60 years or over (as a proxy for an at-risk population). To be considered well-designed, a trial must also measure an appropriate primary outcome–a clinical primary outcome in the case of trials aimed at treating COVID-19, or the presence of laboratory-confirmed SARS-CoV-2 infection for trials testing a preventive measure.

**Feasibility of patient-participant recruitment.** We assessed timeliness and success of patient-participant recruitment for each trial in our cohort. A single trial was considered non-feasible if it met any of the following criteria: i) trial status was "terminated" or "suspended" and reason for stopping contained a rationale unrelated to trial efficacy, safety or the progression of science; ii) trial status was "completed" or "active, not recruiting" and final enrollment was less than 85% of the anticipated enrollment reported in the trial registration at the time of

trial launch (given concerns for compromised statistical power for the primary outcome when recruitment is below the stated threshold (based on previously published methods [17]); or, iii) trial status was "recruiting" or "enrolling by invitation" and the recruitment period had been extended to at least twice as long as the anticipated length in the version of ClinicalTrials.gov registration record at the time of trial start.

## Data analysis

We report the overall proportion of trials meeting all three criteria of informativeness (potential redundancy, design quality and feasibility of patient-participant recruitment) as well as the proportion meeting each of our three criteria. We performed a stratified analysis of the proportion of i) non-redundant; ii) well-designed; and iii) feasible trials by sponsor (industry versus non-industry), trial country location (USA versus non-USA), trial type (treatment versus prevention) and number of trial centers (single center versus multicenter). Ninety-five percent confidence intervals were calculated for the difference between two proportions using the prop.test package in R [18]. All tests were 2-tailed. We followed the Strengthening the Reporting of Observational Studies in Epidemiology (STROBE) reporting guidelines for cohort studies (S1 Checklist) [19].

## Tools and data synthesis

We performed data extraction using Numbat Systematic Review Manager v. 2.11 (RRID: SCR_019207) [20]. All analyses were performed using R version 3.6.3 [21]. We retrieved historical versions of ClinicalTrials.gov using R package 'cthist' (RRID:SCR_019229).

Our study was not subject to Institutional Review Board/Ethics Committee approval, as it relies on publicly accessible data and did not involve interaction with research participants. The study protocol was prospectively registered on Open Science Framework [22]. We listed the deviations from the protocol in S6 File. The code [23] and data sets [22] used in this analysis are available online.

## Results

We included 500 interventional SARS-CoV-2 treatment and prevention efficacy trials (see S1 Fig for Flow Diagram). The number of trials was arrived at by chance and was not predetermined. The majority (58.0%) of trials in our cohort were Phase 2 trials; 84.6% were randomized; 84.8% were directed at the treatment of SARS-CoV-2. Study status at 6 months since trial start was "Completed" in 54 of 500 trials (10.8%) and "Recruiting" in 67.0% (Tables 1, S2 and S3). Median anticipated enrollment per trial (based on the enrollment stated in the last registration record prior to trial start) was 180 patient-participants (range 5–15000 patient-participants; interquartile range (IQR) 60–437). Median actual patient-participant enrollment at the 6-month mark, for those trials that provided actual enrollment numbers, was 129 (range 0–4891 patient-participants; IQR 32–320).

Less than one third (29.9%, 95% CI 23.7–36.9%) of the 194 trials eligible for assessment of all 3 criteria were deemed informative. Nineteen trials were classified as potentially redundant (4.1%), of which 10 investigated convalescent plasma and a further 4 investigated hydroxychloroquine. Sixty-three trials (13.6%) differed only by primary outcome. In our *post hoc* analysis, 81.9% (380 of 464 trials) were similar with respect to trial type, regimen, phase and patient-participant characteristics.

Of the subset of 210 Phase 2/3 and Phase 3 trials in our cohort, 92 (43.8%) met our criteria for trial design quality [20] (Fig 1; Table 2). The proportion of feasible trials in our cohort was 77.4% (387 of 500 trials); 113 trials were non-feasible. Of these, 12 were "Suspended" or

**Table 1. Characteristics of trial cohort.**

| Category | Number of Trials (N = 500) | Percent Total (%) | Median (IQR) Anticipated Enrollment[a] | Median (IQR) Actual Enrollment[b] |
|---|---|---|---|---|
| Trial Phase | | | | |
| Phase 1/2 & Phase 2 | 290 | 58.0 | 100 (40–200) | 60 (25–152) |
| Phase 2/3 & Phase 3 | 210 | 42.0 | 400 (183–1000) | 241 (95–494) |
| Randomization | | | | |
| Randomized | 423 | 84.6 | 200 (82–482) | 142 (53–357) |
| Non-Randomized | 30 | 6.0 | 73 (30–248) | 38 (20–102) |
| NA[c] | 47 | 9.4 | 37 (20–100) | 27 (10–50) |
| Trial Status[d] | | | | |
| Completed | 54 | 10.8 | 100 (46–396) | 100 (40–387) |
| Terminated | 16 | 3.2 | 265 (150–464) | 62 (7–127) |
| Active, Not Recruiting | 71 | 14.2 | 240 (68–500) | 177 (55–442) |
| Recruiting | 335 | 67.0 | 152 (60–410) | 143 (26–230) |
| Enrolling by Invitation | 11 | 2.2 | 128 (56–400) | 72 (51–152) |
| Suspended | 13 | 2.6 | 308 (200–600) | 27 (5–71) |
| Trial Type | | | | |
| Treatment Trial | 424 | 84.8 | 130 (60–333) | 100 (30–233) |
| Prevention Trial | 66 | 13.2 | 672 (206–1729) | 554 (75–1346) |
| Treatment & Prevention | 10 | 2.0 | 782 (250–1500) | 741 (166–1557) |
| Sponsorship | | | | |
| Industry Sponsor | 112 | 22.4 | 195 (82–400) | 187 (84–413) |
| Non-Industry Sponsor | 388 | 77.6 | 177 (60–455) | 100 (27–269) |
| Country Location | | | | |
| USA Trial | 179 | 35.8 | 200 (60–460) | 95 (24–243) |
| Non-USA Trial | 321 | 64.2 | 165 (60–426) | 121 (39–324) |
| Number of Centers | | | | |
| Single Center | 198 | 39.6 | 100 (37–290) | 60 (20–213) |
| Multicenter | 302 | 60.4 | 226 (100–500) | 143 (53–401) |

a) Anticipated enrollment in the first registration record after trial start.

b) At the 6-month mark, for the subset of trials which provide actual enrollment information.

c) NA–Information not available in the ClinicalTrials.gov registration record.

d) Trial Status at the 6-month mark since trial start.

"Terminated "for a reason unrelated to efficacy, safety or the progression of science; 20 trials were "Active, not recruiting" or completed but failed to enrol at least 85% of their target patient-participant enrollment (S2 Fig); 81 trials still "Recruiting" had exceeded at least two times the intended recruitment period (S3 Fig).

## Discussion

Prior studies have examined the COVID-19 trial landscape, evaluating trial design quality [24,25], choice of outcome [26], and presenting descriptive statistics on COVID-19 trials characteristics [2,5,6,8–11]. This is the first study to assess the prevalence of informative COVID-19 clinical trials. In our analysis, 29.9% of early COVID-19 trials registered on ClinicalTrials.gov met our 3 criteria for informativeness. Many (56.2%) did not use rigorous design, based on assessment of randomization, control group, blinding, primary outcome, and inclusion of an at-risk population. Of these, the greatest number (110 of 210 trials, 52.4%) did not demonstrate adequate blinding. Lack of blinding among COVID-19 trials has been highlighted in

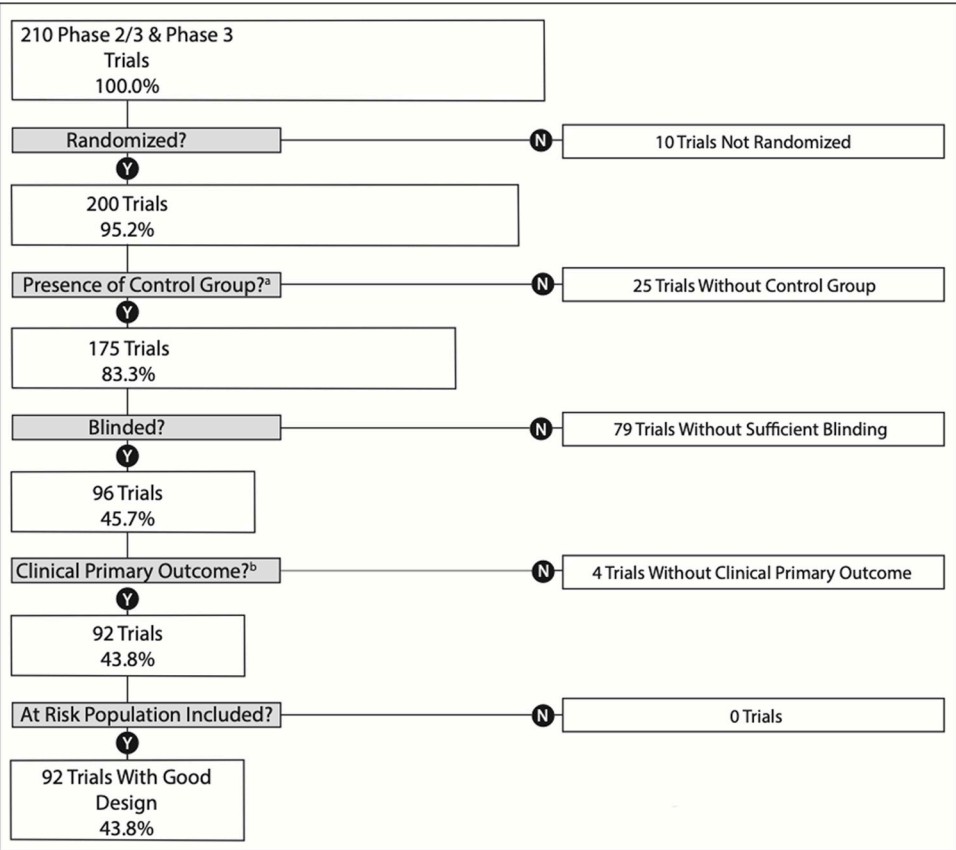

**Fig 1. Flow diagram for trial design quality of Phase 2/3 and Phase 3 SARS-CoV-2 trials.** a) Refers to trial that is either placebo-controlled or has a standard of care comparator arm. b) Refers to a treatment trial with a clinical primary outcome or a prevention trial with either a clinical primary outcome or laboratory-confirmed SARS-CoV-2.

several recent analyses [2,5,6,9,10] and may reflect the challenges of trial conduct in pandemic circumstances, in which significant research infrastructure and oversight is required to implement and maintain blinding. Yet, deficits in trial design were not uniform. Our stratified results (Table 3) demonstrated that trials with at least one center in the USA, in addition to

**Table 2. Evaluation of design quality of trials meant to inform clinical practice.**

| Category | Number of Trials (N = 210) | Percent Total (%) |
|---|---|---|
| Randomized | 200 | 95.2 |
| Placebo-Controlled | 179 | 85.2 |
| Blinded[a] | 100 | 47.6 |
| Clinical Primary Outcome[b] | 203 | 96.7 |
| Includes at Risk Population[c] | 208 | 99.0 |
| **Trials Meeting all 5 Criteria** | **92** | **43.8** |

a) Refers to trials that were at a minimum double-blinded.

b) Treatment trials required a primary clinical outcome; prevention trials required either a primary clinical outcome or laboratory-confirmed SARS-CoV-2.

c) We defined an at risk population as a trial including participants aged $\geq$ 60.

**Table 3. Stratified analysis of redundancy, design, trial feasibility and informativeness by sponsor, country location, trial type, number of trial centers.**

| Informative Condition | Yes (%) | No (%) | \| Difference \| (95% CI) |
|---|---|---|---|
| Non-Redundant | | | |
| Industry Sponsored | 99.1 | 94.9 | 4.1 (0.6–7.6) |
| USA Trial | 95.8 | 96.0 | 0.2 (-3.8–4.2) |
| Treatment Trial | 95.9 | 95.5 | 0.4 (-5.4–6.3) |
| Multicenter Trial | 94.7 | 97.8 | 3.1 (-0.8–6.9) |
| Good Design | | | |
| Industry Sponsored | 73.9 | 35.4 | 38.5 (22.5–54.6) |
| USA Trial | 72.4 | 32.9 | 39.5 (24.6–54.4) |
| Treatment Trial | 39.2 | 62.9 | 23.7 (4.4–43.0) |
| Multicenter Trial | 48.7 | 31.0 | 17.6 (2.1–33.2) |
| Feasible | | | |
| Industry Sponsored | 71.4 | 79.1 | 7.7 (-2.2–17.6) |
| USA Trial | 69.8 | 81.6 | 11.8 (3.4–20.2) |
| Treatment Trial | 78.3 | 71.2 | 7.1 (-5.4–19.6) |
| Multicenter Trial | 73.2 | 83.8 | 10.7 (3.1–18.2) |
| Informative[a] | | | |
| Industry Sponsored | 52.2 | 23.0 | 29.2 (11.8–46.6) |
| USA Trial | 40.7 | 25.7 | 15.0 (-1.2–31.3) |
| Treatment Trial | 28.4 | 31.4 | 3.0 (-15.6–21.7) |
| Multicenter Trial | 30.1 | 29.2 | 1.0 (-14.9–16.8) |

a) Informative trials are those that meet all 3 informativeness criteria.

trials with industry sponsorship, SARS-CoV-2 prevention trials and multicenter trials, demonstrated a greater proportion of well-designed trials than their counterparts.

Despite elevated SARS-CoV-2 cases, many trials (22.6% (113 of 500 trials)) were unable to adequately and expeditiously complete patient-participant recruitment. This estimate is in keeping with other studies in which close to one third of COVID-19 trials registered on ClinicalTrials.gov or on the World Health Organization International Clinical Trials Registry Platform stopped before attaining 75% accrual [27]. In some cases failure to reach recruitment goals can be explained by decreasing case counts in the setting of rapid suppression of a COVID outbreak. For example, early stoppage of a Remdesivir multicenter randomized controlled trial after recruitment of 237 of 453 patient-participants in Wuhan, China, resulted in an underpowered trial with inconclusive results [28,29]. This has also been seen in other settings, such as in the 2014–2016 Ebola outbreak [30]. However, infeasible recruitment targets, despite high case counts, have also been documented during the COVID-19 pandemic [31]. Trial feasibility may be particularly challenging in the fractured US healthcare setting due to inter-trial competition in patient-participant recruitment, as supported by our stratified analysis in which non-USA trials were significantly more likely to be feasible than USA trials.

Lack of coordination and trial prioritization, resulting in a high level of multiplicity in investigated interventions, is a contributing factor to infeasible patient-participant recruitment. Concern about trial redundancy has been brought up frequently during the COVID-19 pandemic [1,2,4,5]. In our study, only 4.1% of trials were deemed potentially redundant, of which 4 investigated hydroxychloroquine and 10 investigated the efficacy of convalescent plasma. Our categorization of trials as potentially redundant involved matching of trial phase, type of trial (treatment versus prevention), patient-participant characteristics, regimen,

comparator and primary outcome. It differs from other assessments of SARS-CoV-2 trial duplication, in which trial intervention has been the main focus of assessment [2]. While a low proportion of potentially redundant trials may be seen as an encouraging result, deeper examination reveals that sixty-three trials (13.6%) assessed for potential redundancy differed only by the choice of primary outcome, with endpoints often demonstrating small deviations from comparator trials, of questionable clinical relevance. For instance, some trials expressed the primary endpoint as a function of time e.g., time to death, whereas in others as a rate e.g., case fatality rate. Our *post hoc* analysis of trial similarity, which evaluated trial type, regimen, phase and patient-participant characteristics, revealed that 81.9% of trials were similar, reflecting the extent to which early clinical trials during the COVID-19 pandemic pursued comparable study designs.

Replication in research is important to clarify study results. However, lack of research coordination and harmonization of primary outcome endpoints during the COVID-19 pandemic [2,4,32,33] can thwart efforts to clarify net effects through meta-analyses. This is particularly relevant in the setting of multiple small trials of specific interventions, where the probability is elevated that at least one trial produces a positive result by chance alone [2,5]. Prospective meta-analyses (PMA), which encourage harmonization of core outcomes and draw on individual participant data, can help clarify treatment effects and reduce research waste [34]. In this way, individually underpowered studies can help address questions of significant clinical importance. Although successfully employed in other medical settings [35,36], PMAs were unfortunately not widely deployed in the early COVID-19 pandemic.

Concerns regarding research waste predated the pandemic [37–43] but intensified in the setting of this international public health crisis. Our results support arguments for devising coordinated research plans in advance of public health emergencies [44], and evaluating and prioritizing trials at institutional [45,46], state and national levels [47]. The success of multi-center national platform trials, such as RECOVERY, in the United Kingdom, in both recruiting patient-participants (over 45580 have been enrolled as of December 9 2021, https://www.recoverytrial.net) and in generating practice-changing evidence, speaks to the promise of national research prioritization [48]. Additional strategies to improve pandemic preparedness include: i) promotion of individual participant data sharing platforms to capitalize on data generated, even from small trials [49]; ii) prioritization of adaptive master protocol trials investigating promising interventions [44,49]; and, iii) increased research collaboration, in the model of the Coalition for Epidemic Preparedness Innovations (CEPI). In our stratified analysis, industry-sponsored trials were significantly more likely to meet all 3 informativeness criteria than non-industry sponsored trials (Table 3). This suggests that academic researchers require more institutional support, as well as assistance from research consortia and funding bodies to produce informative results.

## Limitations

First, we limited our assessment to 3 aspects of trial informativeness–potential redundancy, design quality and feasibility of patient-participant recruitment. Other aspects of informativeness, such as integrity and reporting, were not evaluated in our study, as they cannot be assessed without access to final trial results (430 of 500 trials, 86.0% had not yet completed or terminated at the end of our 6-month follow-up period). A follow-up study evaluating data 24 months after trial launch would enable a comprehensive assessment of trial informativeness, and thus represents an area for future research. Second, we used proxy measures of informativeness, which are imperfect. For example, we adopted strict criteria for potential redundancy, resulting in only 19 trials labelled potentially redundant, many of which differed based on

primary outcome alone. Our *post hoc* analysis resulted in over eighty percent of trials deemed similar, based on assessment of trial type, regimen, phase and patient-participant characteristics. These two results (4.1% and 81.9%) can be viewed as lower and upper bounds for the proportion of redundant trials. Missing from our assessment was an evaluation of the availability and quality (as assessed by GRADE [50]) of pre-existent evidence of intervention efficacy which may render subsequent trials redundant. We also did not assess the extent to which individual participant data were made publicly available (for example, through the Vivli platform [51]), and subsequently incorporated into meta-analyses. Our redundancy evaluation should thus be interpreted with caution and future research will be required to provide a more precise estimate. Our assessment of trial design quality, as guided by the May 2020 FDA guidance document [16], required that all trials be, at a minimum, double-blinded. We acknowledge that this may unfairly penalize the small minority of trials evaluating interventions in which double-blinding is not practicable. In addition, our assessment of the inclusion of at-risk populations was limited only to age. We did not assess whether the study included a population with other risk factors such as comorbidities. However, no trials failed our design criteria based on failure to include an at-risk population. Third, our assessment of the informativeness of COVID-19 trials depends on the accuracy of ClinicalTrials.gov registration records. Fourth, our findings may not be generalizable to all COVID-19 interventional clinical trials. For example, public health behavioural interventions are frequently labelled as "Phase NA" and would therefore not be included in our findings.

## Conclusions

The SARS-CoV-2 pandemic was met with a vigorous response from clinical researchers. However, less than one third of early COVID-19 trials registered on ClinicalTrials.gov met our 3 criteria for informativeness. Shortcomings in trial design, recruitment feasibility and redundancy reflect longstanding vulnerabilities in the clinical research enterprise that were magnified by the urgency of a pandemic. Much knowledge has been gained since the first six months of the COVID-19 pandemic, both in terms of effective measures aimed at treatment and prevention of the virus, but also with respect to the conduct of informative clinical research. The task ahead will be for investigators, research institutions, sponsors and regulators alike to take stock of lessons learned and devise solutions to benefit the global research enterprise as we move forward.

## Supporting information

**S1 Checklist. STROBE statement—Checklist of items that should be included in reports of *cohort studies*.**
(DOCX)

**S1 Fig. Flow diagram of trial inclusion/exclusion.**
(DOCX)

**S2 Fig. Ratio of actual to estimated number of patients enrolled.**
(DOCX)

**S3 Fig. Ratio of actual to estimated recruitment length.**
(DOCX)

**S1 Table. Inter-rater agreement.**
(DOCX)

**S2 Table. Additional characteristics of trial cohort.**
(DOCX)

**S3 Table. Range of anticipated and actual enrollment.**
(DOCX)

**S1 File. Trial inclusion and exclusion criteria.**
(DOCX)

**S2 File. ClinicalTrials.gov search criteria.**
(DOCX)

**S3 File. Data downloaded from ClinicalTrials.gov.**
(DOCX)

**S4 File. Additional data points.**
(DOCX)

**S5 File. Assessment of trial redundancy.**
(DOCX)

**S6 File. Protocol deviations.**
(DOCX)

## Acknowledgments

We thank Lucja Zabrowska for her important contribution to the data extraction for this project and Maciej Polak for statistical consultancy.

## Author Contributions

**Conceptualization:** Nora Hutchinson, Katarzyna Klas, Benjamin G. Carlisle, Marcin Waligora.

**Data curation:** Katarzyna Klas.

**Formal analysis:** Nora Hutchinson, Benjamin G. Carlisle.

**Funding acquisition:** Marcin Waligora.

**Investigation:** Nora Hutchinson, Katarzyna Klas, Benjamin G. Carlisle.

**Methodology:** Nora Hutchinson, Katarzyna Klas, Benjamin G. Carlisle, Jonathan Kimmelman, Marcin Waligora.

**Project administration:** Nora Hutchinson, Marcin Waligora.

**Software:** Nora Hutchinson, Benjamin G. Carlisle.

**Supervision:** Marcin Waligora.

**Visualization:** Nora Hutchinson.

**Writing – original draft:** Nora Hutchinson.

**Writing – review & editing:** Nora Hutchinson, Katarzyna Klas, Benjamin G. Carlisle, Jonathan Kimmelman, Marcin Waligora.

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
