## [Decision Letter · Decision Letter 0]

2 Nov 2021

PONE-D-21-28808How Informative Were Early SARS-CoV-2 Treatment and Prevention Trials? A longitudinal cohort analysis of trials registered on clinicaltrials.govPLOS ONE

Dear Dr. Waligora,

Thank you for submitting your manuscript to PLOS ONE. After careful consideration, we feel that it has merit but does not fully meet PLOS ONE’s publication criteria as it currently stands. Therefore, we invite you to submit a revised version of the manuscript that addresses the points raised during the review process.

We look forward to receiving your revised manuscript.

Kind regards,

Dylan A Mordaunt, MB ChB, FRACP, FAIDH

Academic Editor

PLOS ONE

Journal Requirements:

“Marcin Waligora reports personal fees from Advisory Bioethics Council, Sanofi outside the submitted work. Other authors have declared that no competing interests exist.”

Additional Editor Comments:

Thank you for your submission in a format that is considered for publication in PLoS One. The reviewers have included a number of useful recommendations for revision of the manuscript. With reference to the publication for criteria:

1. The study appears to represent the results of original meta-research. If there have been similar studies before or since, it would be worth commenting on these for completeness.

2. Results do not appear to have been published elsewhere.

3. Experiments, statistics, and other analyses require some work. These are detailed by reviewer 1, 3 and 4.

4. Conclusions are presented in an appropriate fashion and are supported by the data. The reviewers don't comment on the conclusions, however I agree with the comment on expanding on narrative synthesis in the discussion.

5. The article is presented in an intelligible fashion and is written in standard English.

6. The research meets all applicable standards for the ethics of experimentation and research integrity.

7. The article adheres to appropriate reporting guidelines and community standards for data availability. Without exhaustively detailing them here, it would be helpful to have the manuscript follow standardised reporting structures- although there may not be one specific to this study type, ones that relate to systematic reviews of observational studies or observational studies more generally, would be helpful such as PRISMA and AMSTAR-2. Specific features such as whether the protocol was prospectively registered, where it was registered, the detail of how duplicate assessment occurred etc., should be included. In that specific example, it may be worth addressing in the response if the protocol wasn't prospectively addressed, so that it could be taken into account in future meta-research.

Reviewers' comments:

Reviewer's Responses to Questions

**Comments to the Author**

1. Is the manuscript technically sound, and do the data support the conclusions?

Reviewer #1: Yes

Reviewer #2: Yes

Reviewer #3: Partly

Reviewer #4: Partly

2. Has the statistical analysis been performed appropriately and rigorously? 

Reviewer #1: No

Reviewer #2: Yes

Reviewer #3: Yes

Reviewer #4: Yes

3. Have the authors made all data underlying the findings in their manuscript fully available?

Reviewer #1: Yes

Reviewer #2: Yes

Reviewer #3: Yes

Reviewer #4: Yes

4. Is the manuscript presented in an intelligible fashion and written in standard English?

Reviewer #1: Yes

Reviewer #2: Yes

Reviewer #3: Yes

Reviewer #4: Yes

5. Review Comments to the Author

Reviewer #1: General comment

This seems to me a very important study worthy of publication. The paper underlines how important it is that trials are conducted by appropriately supported centres with experience of conducting trials. However, although the study compares the properties between groups within features, for example, Multicentre v Single Centre, the difference between them should be presented together with the 95% CI of that difference (see below).

Specific comments

1. Page 10, Table 1: Probably more informative if the row Phase 2c is spilt into two rows one for Phase 1 and a second for Phase 2. Also confusing as how in row Phase 3d the Phase 2 components mentioned differ from those in row Phase 2c. There is some confusion here and consequently on Page 10, line 7, where the authors state: “of 210 Phase 2/3 and Phase 3 trials”, it remains unclear as to what this group actually comprises.

2. Page 10, Table 1 think the actual range (minimum and maximum values), rather than IQR of the anticipated recruitment and actual enrolment, would be much more informative. Also, in the written text above on Page 9.

3. Page 12, Table 2 footnote c) “Age of participants � 60; the two trials not including participants � 60 years of age included healthy adults without any additional factors putting them at greater risk for severe SARS-CoV-2 disease”. It was not at all clear to me what is meant by this statement.

4. Page 12, Figure 1: Again, the confusion remains between Phase 2/3 and Phase 3.

5. Page 8, line 5 from bottom. I am not sure that statistical significance tests are required (see below). However, it is better to interpret the actual p-value rather than state “We defined p < 0.05 as statistically significant”. I suggest omit this phrase.

6. Page 14, Table 3: It would be useful to quote the statistical package used for the calculation of the exact CIs.

7. Page 14, Table 3: Too much precision clouds the message. I suggest replacing, for example, 99.11 (95.13 – 99.98) by 99.1 (95.1 – 100) although quoting these CIs is unnecessary. However, what would be useful is to quote their difference 99.11 − 95.36 = 3.75 with its 95%CI 1.02 to 6.47 and the corresponding p-value = 0.0070. I suggest a better format for Table 3 might be:

Type of Trial Yes (%) No (%) Difference 95%CI p-value

Industry Sponsor 99.1 95.4 3.8 1.0 to 6.5 0.068

USA 96.1 96.2 −0.2 −3.7 to +3.3 0.92

Therapeutic 96.2 95.5 0.8 −4.6 to +6.1 0.76

Multicentre 95.0 98.0 −3.0 −6.1 to +0.2 0.092

Technical note

When comparing differences between proportions which involve any values close to 100% (and or 0%) cause technical problems. Thus, there are several approaches to these calculations and these may give differing p-values. The Exact method is one the authors refer to in their Table 3 which seems entirely appropriate. However, my calculations above have used the statistical package Stata to obtain the p-values which differ somewhat from those of the author. As I indicate above, I am not sure it is necessary to calculate the p-values. Interpretation should focus on the magnitude of the differences and their CIs.

Reviewer #2: Interesting, well researched and timely manuscript. Clearly written. Conducive to a follow up paper in 12-18 months (6 months is a relatively short period of time), to see whether a longer snapshot e.g 12 or 24 month timeframe changes the results/conclusions.

Reviewer #3: This article described an analysis on early trials of COVID-19 and their informativeness. I would like to congratulate the authors an important piece of work describing potential shortcomings in trial design, recruitment, and potential redundancy. The protocol for this work has been prospectively registered, and there is a clear list of protocol deviations. The manuscript is well-written and easy to follow. I have a few comments for the authors to consider.

Abstract:

The abstract could be refined to be more informative as a standalone. It would be helpful to include some more specific information on how the three criteria on informativeness were defined, and how the cohort was created (eligibility criteria? Random selection of trials or own trials?).

Methods: Did the authors adhere to a reporting checklist (e.g. STROBE or PRISMA)? It would be good to include this checklist as a supplement.

Eligibility criteria: The inclusion and exclusion criteria should be listed in the manuscript, and not only provided in a supplement. The reasoning behind the choice of eligibility criteria is unclear, and should be elaborated. I wonder if some of the choices impede generalizability of results. For instance, study phase is a criterion that is usually only filled in for drug trials on trial registries, other trials often chose the option ‘not applicable’. I wonder if by restricting this analysis to certain phase trials, information on other trials was lost? In addition, restricting to Phase 1/2-3 & only trials testing for efficacy may exclude non-drug interventions such as public health messaging trials. Why were behavioral interventions, dietary supplement and Chinese medicine trials excluded?

Search string: It is unclear how trials were identified on ClinicalTrials.gov. Were filters used (e.g. COVID-19 or Phase filters?). Or did the authors include all registrations within a time frame?

Trial screening and coding of outcomes: What was the agreement between screeners? How were disagreements resolved? Were informativeness measures also assessed by two screeners (this is implied but not explicit)? How was the agreement?

Informativeness concepts: The authors refer the reader to information on ‘Informativeness articulated elsewere’ (p.6), to understand the assessed concepts. Since this is a core construct that is required to be understood to understand this paper, I would recommend introducing these concepts and what they mean in detail in the introduction.

Redundancy: I have some reservations about the assessment of this concept. Replication in research is crucial, and often trials (and particularly early trials) do not have sufficient sample size to conclusively answer a research question.

A trial is only redundant, if high certainty evidence exists that an intervention is effective or not effective (as evaluated by GRADE). This does not seem to have been assessed in this case. If certainty of evidence is low, additional replication trials are crucial to ensure early findings were not purely contextual or chance findings (and thus, they are not redundant in this case). For this reason, I would interpret this criterion very carefully.

A slightly different primary outcome does not necessarily make a trial non-redundant. In fact, as the authors point out in the discussion, it may be better if two trials collect the same outcomes so they can be combined in meta-analysis. The analysis looking at the numbers of trials labelled as redundant when disregarding the primary outcome is important, it may be worthwhile presenting this analysis more prominently.

Design quality:

Trial design was only analysed for Phase 2/3 and Phase 3 trials – but trial design is also important for earlier phase trials (albeit criteria may be different)?

‘We considered a trial to be well-designed if it was randomized, placebo-controlled (with appropriate standard of care in all arms), double-blinded and included participants aged 60 years or over (as a proxy for an at-risk population)’ What if a trial had an active control? Would that not be considered well-designed?

It would have been good to look at each trial design criterion separately in each trial (and not just the ones that satisfied previous requirements), to get an assessment of how well each design feature was fulfilled in those trials.

Feasibility of Patient-Participant Recruitment:

How would a trial that stopped early for effectiveness be assessed here? Also, from our experience of managing a registry, many registrants do not update their registration records even if they have finished recruiting, thus, a trial may have long finished recruitment and still be listed as ‘recruiting’. Do the authors have information on how many of the trials have updated their records?

Table 1: Characteristics of trial cohort. If possible, it would be great to include some additional information on the trials, such as target sample sizes and included populations.

Discussion: I would be interested in a more in-depth discussion of what needs to change on a structural level in future to improve trial informativeness, particularly in the context of health emergencies.

Reviewer #4: 1. Was the sample of exactly 500 arrived at purely by chance? If so, please make it clear that this was not a predetermined number.

2. It would be valuable to include a checklist of items according to the STROBE guidelines https://www.strobe-statement.org/checklists/and STROBE-checklist-v4-combined-PlosMedicine.pdf, with corresponding page numbers to indicate where each item is addressed,

3. I have a big problem with the definition of redundancy as the presence of another trial of the same phase, type of trial (SARS-CoV-2 prevention versus treatment), patient-participant characteristics (including location of care, disease severity and age of trial participants), regimen (including interventions used in combination in a single arm), comparator arm(s) and primary outcome (evaluating primary outcome domain and specific measurement, based on framework from ref 13. This excludes the highly desirable situation when multiple investigators who have obtained funding from a funding agency for a single smaller trial agree to undertake a prospective meta-analysis of individual participant data, as in the NeOProM Collaboration of RCTs of oxygen targeting in preterm newborns (Askie et al JAMA 2018) and (Askie et al Pediatric Obesity 2020 https://onlinelibrary.wiley.com/doi/abs/10.1111/ijpo.12618 and other next-generation syntheses of similar trials to enhance power (see Seidler et al Guide to Prospective Meta-Analysis, BMJ 2019).

4. In view of 3, it is essential in the Discussion to acknowledge that (i) even individually underpowered trials can make a valuable contribution in addressing critically important questions regarding mortality if included in individual participant data meta-analyses and(ii) inability to assess how often this was happening is a major limitation of this study.

I would recommend the manuscript be substantially revised and resubmitted.

Thank you for the opportunity to review this important work.

6. PLOS authors have the option to publish the peer review history of their article (what does this mean?). If published, this will include your full peer review and any attached files.

Reviewer #1: **Yes: **David Machin

Reviewer #2: No

Reviewer #3: **Yes: **Anna Lene Seidler

Reviewer #4: **Yes: **William Odita Tarnow-Mordi

---

## [Author Response · Author response to Decision Letter 0]

17 Dec 2021

Dear Editors and Referees:

We thank the referees and editors for their careful and constructive assessment of our manuscript. We are grateful for the opportunity to revise and resubmit our manuscript. Responses to all comments are appended below.

Editors’ Comments:

1. The study appears to represent the results of original meta-research. If there have been similar studies before or since, it would be worth commenting on these for completeness.

Yes, this manuscript does represent original meta-research. We have added the following sentence to page 14, lines 247-249, with additional citations of recently published meta-research, not included in our original submission: “Prior studies have examined the COVID-19 trial landscape, evaluating trial design quality, choice of outcome, and presenting descriptive statistics on COVID-19 trials characteristics.”

2. Results do not appear to have been published elsewhere.

Our manuscript is available on the medRxiv preprint server. Results have not been published elsewhere.

3. Experiments, statistics, and other analyses require some work. These are detailed by reviewer 1, 3 and 4.

Thank you. We have altered our analyses, as detailed below.

4. Conclusions are presented in an appropriate fashion and are supported by the data. The reviewers don't comment on the conclusions, however I agree with the comment on expanding on narrative synthesis in the discussion.

We have expanded on the discussion, addressing the following points:

-We call attention to our post hoc assessment of trial similarity (as recommended by Reviewer # 3): page 17, lines 302-305: “Our post hoc analysis of trial similarity, which evaluated trial type, regimen, phase and patient-participant characteristics, revealed that 81.9% of trials were similar, reflecting the extent to which early clinical trials during the COVID-19 pandemic pursued comparable study designs.”

-We discuss prospective meta-analyses (as recommended by Reviewer # 4): pages 17-18, Lines 312-317: “Prospective meta-analyses (PMA), which encourage harmonization of core outcomes and draw on individual participant data, can help clarify treatment effects and reduce research waste. In this way, individually underpowered studies can help address questions of significant clinical importance. Although successfully employed in other medical settings, PMAs were unfortunately not widely deployed in the early COVID-19 pandemic. 

-We suggest additional strategies to improve pandemic preparedness (as recommended by Reviewer # 3): page 18, lines 326-331): “Additional strategies to improve pandemic preparedness include: i) promotion of individual participant data sharing platforms to capitalize on data generated, even from small trials;43 ii) prioritization of adaptive master protocol trials investigating promising interventions;38,43 and, iii) increased research collaboration, in the model of the Coalition for Epidemic Preparedness Innovations (CEPI).”

5. The article is presented in an intelligible fashion and is written in standard English.

Thank you.

6. The research meets all applicable standards for the ethics of experimentation and research integrity.

Thank you.

7. The article adheres to appropriate reporting guidelines and community standards for data availability. Without exhaustively detailing them here, it would be helpful to have the manuscript follow standardised reporting structures- although there may not be one specific to this study type, ones that relate to systematic reviews of observational studies or observational studies more generally, would be helpful such as PRISMA and AMSTAR-2. Specific features such as whether the protocol was prospectively registered, where it was registered, the detail of how duplicate assessment occurred etc., should be included. In that specific example, it may be worth addressing in the response if the protocol wasn't prospectively addressed, so that it could be taken into account in future meta-research.

Our study protocol was prospectively registered with the Open Science Framework, as indicated in the Abstract on Page 2, Lines 38-40: “The study protocol was prospectively registered with the Open Science Framework (https://osf.io/fp726/)” and in the Methods section on Page 9, lines 191-192.

We have added a STROBE checklist for cohort studies, as indicated on page 9, lines 178-180: “We followed the Strengthening the Reporting of Observational Studies in Epidemiology (STROBE) reporting guidelines for cohort studies.” The checklist is in the supplementary appendix (S1 Checklist). 

Reviewer # 1 Comments:

This seems to me a very important study worthy of publication. The paper underlines how important it is that trials are conducted by appropriately supported centres with experience of conducting trials. However, although the study compares the properties between groups within features, for example, Multicentre v Single Centre, the difference between them should be presented together with the 95% CI of that difference (see below).

Thank you. We have made the suggested changes to our analysis, as described under question # 7 below.

1. Page 10, Table 1: Probably more informative if the row Phase 2c is spilt into two rows one for Phase 1 and a second for Phase 2. Also confusing as how in row Phase 3d the Phase 2 components mentioned differ from those in row Phase 2c. There is some confusion here and consequently on Page 10, line 7, where the authors state: “of 210 Phase 2/3 and Phase 3 trials”, it remains unclear as to what this group actually comprises.

We included the following categories of trials in our cohort: Phase 1/2, Phase 2, Phase 2/3 and Phase 3 trials. We did not include any Phase 1 trials in our cohort. 

We classified trials that encompass two phases, such as so-called seamless trials: Phase 1/2 trials and Phase 2/3 trials based on the higher phase. Therefore, Phase 1/2 trials were classified with Phase 2 trials and Phase 2/3 trials were classified with Phase 3 trials. 

We have updated Table 1 for clarity in the following way: Column 1, Row 2 now reads: “Phase 1/2 & Phase 2”; Column 1, Row 3 now reads: “Phase 2/3 & Phase 3.” 

We did not update the following sentence: “Of the subset of 210 Phase 2/3 and Phase 3 trials…” as Phase 2/3 trials represent a specific type of trial ("seamless trial”) that encompasses both Phase 2 and Phase 3 within a single trial. 

2. Page 10, Table 1 think the actual range (minimum and maximum values), rather than IQR of the anticipated recruitment and actual enrolment, would be much more informative. Also, in the written text above on Page 9.

Our goal in presenting the interquartile range was to provide an estimate of the variance. Therefore, we have kept the IQR in Table 2, but have added the range in S3 Table. We also added the range for the median anticipated enrollment per trial and actual enrollment per trial on page 10, in lines 203 and 205 respectively.

3. Page 12, Table 2 footnote c) “Age of participants � 60; the two trials not including participants � 60 years of age included healthy adults without any additional factors putting them at greater risk for severe SARS-CoV-2 disease”. It was not at all clear to me what is meant by this statement.

We agree that this sentence was confusing and have removed it from the manuscript. 

4. Page 12, Figure 1: Again, the confusion remains between Phase 2/3 and Phase 3.

As described above, Phase 2/3 and Phase 3 trials both represent distinct types of trials, the former combining Phase 2 and Phase 3 within a single trial. We believe that this is accepted terminology and therefore have not changed the title of Figure 1.

5. Page 8, line 5 from bottom. I am not sure that statistical significance tests are required (see below). However, it is better to interpret the actual p-value rather than state “We defined p < 0.05 as statistically significant”. I suggest omit this phrase.

We have removed this sentence from the manuscript. 

6. Page 14, Table 3: It would be useful to quote the statistical package used for the calculation of the exact CIs.

We have modified the following sentence on page 9, lines 177-178: “Ninety-five percent confidence intervals were calculated for the difference between two proportions using the prop.test package in R.”

7. Page 14, Table 3: Too much precision clouds the message. I suggest replacing, for example, 99.11 (95.13 – 99.98) by 99.1 (95.1 – 100) although quoting these CIs is unnecessary. However, what would be useful is to quote their difference 99.11 − 95.36 = 3.75 with its 95%CI 1.02 to 6.47 and the corresponding p-value = 0.0070. I suggest a better format for Table 3 might be:

Type of Trial Yes (%) No (%) Difference 95%CI p-value

Industry Sponsor 99.1 95.4 3.8 1.0 to 6.5 0.068

USA 96.1 96.2 −0.2 −3.7 to +3.3 0.92

Therapeutic 96.2 95.5 0.8 −4.6 to +6.1 0.76

Multicentre 95.0 98.0 −3.0 −6.1 to +0.2 0.092

Technical note

When comparing differences between proportions which involve any values close to 100% (and or 0%) cause technical problems. Thus, there are several approaches to these calculations and these may give differing p-values. The Exact method is one the authors refer to in their Table 3 which seems entirely appropriate. However, my calculations above have used the statistical package Stata to obtain the p-values which differ somewhat from those of the author. As I indicate above, I am not sure it is necessary to calculate the p-values. Interpretation should focus on the magnitude of the differences and their CIs.

Thank you for this helpful suggestion! We have modified our stratified analysis, as presented in Table 3. The format of the columns is as you proposed. We also now present the difference between proportions with a 95% CI for that difference. We removed p-values from the Table. 

Reviewer # 2 Comments:

Interesting, well researched and timely manuscript. Clearly written. Conducive to a follow up paper in 12-18 months (6 months is a relatively short period of time), to see whether a longer snapshot e.g 12 or 24 month timeframe changes the results/conclusions.

Thank you! We agree that a follow-up study would be very interesting. We have added the following sentence in the limitations section of our manuscript to reflect this (page 19, lines 343-345): “A follow-up study evaluating data 24 months after trial launch would enable a comprehensive assessment of trial informativeness, and thus represents an area for future research.”

Reviewer # 3 Comments:

This article described an analysis on early trials of COVID-19 and their informativeness. I would like to congratulate the authors an important piece of work describing potential shortcomings in trial design, recruitment, and potential redundancy. The protocol for this work has been prospectively registered, and there is a clear list of protocol deviations. The manuscript is well-written and easy to follow. I have a few comments for the authors to consider.

1. Abstract:

The abstract could be refined to be more informative as a standalone. It would be helpful to include some more specific information on how the three criteria on informativeness were defined, and how the cohort was created (eligibility criteria? Random selection of trials or own trials?).

Thank you for this suggestion. We have updated the abstract, providing more information about our eligibility criteria: (page 2, lines 29-33): “Based on prespecified eligibility criteria, we created a cohort of Phase 1/2, Phase 2, Phase 2/3 and Phase 3 SARS-CoV-2 treatment and prevention efficacy trials that were initiated from 2020-01-01 to 2020-06-30 using ClinicalTrials.gov registration records. We excluded trials evaluating behavioural interventions and natural products, which are not regulated by the U.S. Food and Drug Administration (FDA).” We also specified that all eligible trials were included in our cohort (page 2, line 42): “We included all 500 eligible trials in our cohort…” 

A more detailed description of the 3 criteria for informativeness was also added to the Methods section of the abstract (page 2, lines 33-38): “We evaluated trials on 3 criteria of informativeness: potential redundancy (comparing trial phase, type, patient-participant characteristics, treatment regimen, comparator arms and primary outcome), trials design (according to the recommendations set-out in the May 2020 FDA guidance document on SARS-CoV-2 treatment and prevention trials) and feasibility of patient-participant recruitment (based on timeliness and success of recruitment).”

2. Methods: Did the authors adhere to a reporting checklist (e.g. STROBE or PRISMA)? It would be good to include this checklist as a supplement.

We have followed the STROBE checklist for cohort studies. We have added the following sentence to the Methods section (page 9, lines 178-180): “We followed the Strengthening the Reporting of Observational Studies in Epidemiology (STROBE) reporting guidelines for cohort studies.” The STROBE checklist is now provided in the supplemental appendix (S1 Checklist).

3. Eligibility criteria: The inclusion and exclusion criteria should be listed in the manuscript, and not only provided in a supplement. The reasoning behind the choice of eligibility criteria is unclear, and should be elaborated. I wonder if some of the choices impede generalizability of results. For instance, study phase is a criterion that is usually only filled in for drug trials on trial registries, other trials often chose the option ‘not applicable’. I wonder if by restricting this analysis to certain phase trials, information on other trials was lost? In addition, restricting to Phase 1/2-3 & only trials testing for efficacy may exclude non-drug interventions such as public health messaging trials. Why were behavioral interventions, dietary supplement and Chinese medicine trials excluded?

As suggested, we expanded the description of eligibility criteria in our manuscript instead of presenting them only in supporting information (page 5, lines 89-97): “Our cohort consisted of interventional SARS-CoV-2 treatment and prevention trials registered on ClinicalTrials.gov with a start date between 2020-01-01 and 2020-06-30. We included “Completed”, “Terminated”, “Suspended”, “Active, not recruiting”, “Enrolling by invitation” and “Recruiting” Phase 1/2, Phase 2, Phase 2/3 and Phase 3 interventional clinical trials testing an efficacy hypothesis in their primary outcome. We included trials evaluating any of the following interventions: drug, biological, surgical, radiotherapy, procedural or device. We excluded trials evaluating behavioural interventions, trials of natural products and Phase 1 trials, all of which have no legal requirement to register on ClinicalTrials.gov.”

When defining the inclusion and exclusion criteria, we wished to focus primarily on clinical trials of interventions required by law to register on ClinicalTrials.gov (based on 42 CFR Part 11). Behavioural interventions, trials of natural products (which includes the majority of traditional Chinese medicine products, which are often herbal products) and Phase 1 trials have no legal requirement to register on ClinicalTrials.gov, and this informed our decision to exclude them. 

We agree that our cohort therefore does not reflect the full breadth of possible COVID-19 treatment and prevention trials. We have thus added the following to the limitations section (page 20, lines 366-368): “Fourth, our findings may not be generalizable to all COVID-19 interventional clinical trials. For example, public health behavioural interventions are frequently labelled as “Phase NA” and would therefore not be included in our findings.”

4. Search string: It is unclear how trials were identified on ClinicalTrials.gov. Were filters used (e.g. COVID-19 or Phase filters?). Or did the authors include all registrations within a time frame?

We have now added our specific search criteria in supplemental S2 File:

“We downloaded clinical trial data directly as a zipped folder of XML files from the web front-end of ClinicalTrials.gov. 

We used the following search criteria:

411 records identified through 12/01/2021:

Condition or disease: “Covid-19”

Study Type: “Interventional Studies”

Trial Status: “Recruiting, “Active, not recruiting,” “Completed,” “Enrolling by invitation,” “Suspended,” “Terminated” 

Phase: Phase 2, Phase 3

Start Date: 01/01/2020 to 05/31/2020

110 records identified through 01/04/2021:

Condition or disease: “Covid-19”

Study Type: “Interventional Studies”

Trial Status: “Recruiting, “Active, not recruiting,” “Completed,” “Enrolling by invitation,” “Suspended,” “Terminated” 

Phase: Phase 2, Phase 3

Start Date: 06/01/2020 to 06/30/2020”

5. Trial screening and coding of outcomes: What was the agreement between screeners? How were disagreements resolved? Were informativeness measures also assessed by two screeners (this is implied but not explicit)? How was the agreement?

We have added S1 Table which provides the inter-rater agreement for all items that required human curation. Coding was independently performed by two individuals and when necessary a third person, an arbiter, was involved. This is now more precisely stated on page 5, lines 98-100: “Trial inclusion and exclusion criteria were independently assessed by two researchers (KK & LZ), with disagreements resolved by an arbiter (NH or MW),” page 6, lines 109-110 and 116-117: “Additional items requiring human curation were independently assessed and coded by two researchers (KK & LZ) … Disagreements were resolved by an arbiter (NH or MW)” and page 7, lines 137-139: “The assessment was independently performed by two raters (NH & KK), with disagreements resolved by an arbiter (MW of BC).”

Informativeness criteria were assessed as follows. First, the assessment of potential redundancy combined an automated assessment of trial phase and several human curated data points (type of trial, patient-participant characteristics, regimen, comparator arms, and primary outcome). The latter elements were assessed by two individuals (KK and LW) with disagreements resolved by a third (NH or MW). The final assessment of potentially redundant trials was independently performed by two assessors (NH and KK), with disagreements resolved by a third (MW or BC). This is further described in supplemental S5 File. Second, assessment of design quality also combined an automated assessment of trial phase, randomization, blinding and age of participants, with human curated data points (presence of placebo or standard of care arm and primary outcome). The latter two elements were assessed by two individuals (KK and LW) with disagreements resolved by a third (NH or MW). Finally, assessment of the feasibility of patient-participant recruitment was automated. 

6. Informativeness concepts: The authors refer the reader to information on ‘Informativeness articulated elsewhere’ (p.6), to understand the assessed concepts. Since this is a core construct that is required to be understood to understand this paper, I would recommend introducing these concepts and what they mean in detail in the introduction.

Thank you for this suggestion. We have now elaborated on the criteria for informativeness in the introduction (page 4, lines 68-73): “For a trial to be informative to clinical practice, it must fulfill five conditions. First, it must ask a clinically important question. Second, it must be designed to provide a clear answer to that question. Third, it must have both a feasible enrollment target and primary completion timeline. Fourth, it must be analyzed in a manner that supports statistically valid inference. Fifth, it must report results in a complete and timely manner.” 

7. Redundancy: I have some reservations about the assessment of this concept. Replication in research is crucial, and often trials (and particularly early trials) do not have sufficient sample size to conclusively answer a research question.

A trial is only redundant, if high certainty evidence exists that an intervention is effective or not effective (as evaluated by GRADE). This does not seem to have been assessed in this case. If certainty of evidence is low, additional replication trials are crucial to ensure early findings were not purely contextual or chance findings (and thus, they are not redundant in this case). For this reason, I would interpret this criterion very carefully.

We agree that our assessment of redundancy is imperfect. We have elaborated on its shortcomings in the limitations section (page 19, lines 351-357): “Missing from our assessment was an evaluation of the availability and quality (as assessed by GRADE) of pre-existent evidence of intervention efficacy which may render subsequent trials redundant. We also did not assess the extent to which individual participant data were made publicly available (for example, through the Vivli platform), and subsequently incorporated into meta-analyses. Our redundancy evaluation should thus be interpreted with caution and future research will be required to provide a more precise estimate.”

A slightly different primary outcome does not necessarily make a trial non-redundant. In fact, as the authors point out in the discussion, it may be better if two trials collect the same outcomes so they can be combined in meta-analysis. The analysis looking at the numbers of trials labelled as redundant when disregarding the primary outcome is important, it may be worthwhile presenting this analysis more prominently.

We have added the following sentence to the discussion to highlight our assessment of trial similarity (page 17, lines 302-305): “Our post hoc analysis of trial similarity, which evaluated trial type, regimen, phase and patient-participant characteristics, revealed that 81.9% of trials were similar, reflecting the extent to which early clinical trials during the COVID-19 pandemic pursued comparable study designs.”

8. Design quality:

Trial design was only analysed for Phase 2/3 and Phase 3 trials – but trial design is also important for earlier phase trials (albeit criteria may be different)?

Yes, we agree that trial design is equally important for earlier phase trials. We chose to confine our assessment of trial design to Phase 2/3 and Phase 3 trials given that we based our assessment of design on the May 2020 FDA guidance document, which focused primarily on later phase trials. 

9. ‘We considered a trial to be well-designed if it was randomized, placebo-controlled (with appropriate standard of care in all arms), double-blinded and included participants aged 60 years or over (as a proxy for an at-risk population)’ What if a trial had an active control? Would that not be considered well-designed?

A trial with an active control arm, or standard of care arm, was also accepted as well designed. The following has been updated for clarify (page 8, lines 148-150): “we considered a trial to be well-designed if it was randomized, placebo-controlled or with a standard of care comparator arm…”

10. It would have been good to look at each trial design criterion separately in each trial (and not just the ones that satisfied previous requirements), to get an assessment of how well each design feature was fulfilled in those trials.

This data is presented in Table 2 (page 12).

11. Feasibility of Patient-Participant Recruitment:

How would a trial that stopped early for effectiveness be assessed here? Also, from our experience of managing a registry, many registrants do not update their registration records even if they have finished recruiting, thus, a trial may have long finished recruitment and still be listed as ‘recruiting’. Do the authors have information on how many of the trials have updated their records?

If a trial was terminated early due to efficacy, this trial was not labelled as infeasible (page 8, lines 158-160): “A single trial was considered non-feasible if it met any of the following criteria: i) trial status was “terminated” or “suspended” and reason for stopping contained a rationale unrelated to trial efficacy, safety or the progression of science…”

Yes, we agree that our analysis was highly dependent on the accuracy of trial records on Clinicaltrials.gov, which may not reflect actual trial status. We have highlighted our reliance on the accuracy of Clinicaltrials.gov records in our limitations section (page 20, lines 364-366): “Third, our assessment of the informativeness of COVID-19 trials depends on the accuracy of ClinicalTrials.gov registration records.”

We evaluated all trials in our cohort at the 6-month mark, to provide equal follow-up time for each clinical trial. A follow-up study, evaluating trial data 24 months after trial launch would be beneficial, and represents an area of future research (page 19, lines 343-345): “A follow-up study evaluating data 24 months after trial launch would enable a comprehensive assessment of trial informativeness, and thus represents an area for future research.”

12. Table 1: Characteristics of trial cohort. If possible, it would be great to include some additional information on the trials, such as target sample sizes and included populations.

We have added S2 Table to our supplement, providing additional information about the included trials including age of participants, location of care and SARS-CoV-2 severity. 

13. Discussion: I would be interested in a more in-depth discussion of what needs to change on a structural level in future to improve trial informativeness, particularly in the context of health emergencies.

Thank you for this suggestion. We have added the following to the discussion (page 18, lines 326-331): “Additional strategies to improve pandemic preparedness include: i) promotion of individual participant data sharing platforms to capitalize on data generated, even from small trials;45 ii) prioritization of adaptive master protocol trials investigating promising interventions;40,45 and, iii) increased research collaboration, in the model of the Coalition for Epidemic Preparedness Innovations (CEPI).”

Reviewer # 4 Comments:

1. Was the sample of exactly 500 arrived at purely by chance? If so, please make it clear that this was not a predetermined number.

Yes, the sample size of 500 trials was arrived at by chance. We have updated our abstract to make clear that we included all eligible trials in our cohort (page 2, line 42): “We included all 500 eligible trials in our cohort…” We have also added the following sentence in our results (page 10, lines 197-198): “The number of trials was arrived at by chance and was not predetermined.”

2. It would be valuable to include a checklist of items according to the STROBE guidelines https://www.strobe-statement.org/checklists/and STROBE-checklist-v4-combined-PlosMedicine.pdf, with corresponding page numbers to indicate where each item is addressed.

We have added a STROBE checklist for cohort studies, as indicated on page 9, lines 178-180: “We followed the Strengthening the Reporting of Observational Studies in Epidemiology (STROBE) reporting guidelines for cohort studies.” The checklist is in the supplementary appendix (S1 Checklist). 

3. I have a big problem with the definition of redundancy as the presence of another trial of the same phase, type of trial (SARS-CoV-2 prevention versus treatment), patient-participant characteristics (including location of care, disease severity and age of trial participants), regimen (including interventions used in combination in a single arm), comparator arm(s) and primary outcome (evaluating primary outcome domain and specific measurement, based on framework from ref 13. This excludes the highly desirable situation when multiple investigators who have obtained funding from a funding agency for a single smaller trial agree to undertake a prospective meta-analysis of individual participant data, as in the NeOProM Collaboration of RCTs of oxygen targeting in preterm newborns (Askie et al JAMA 2018) and (Askie et al Pediatric Obesity 2020 https://onlinelibrary.wiley.com/doi/abs/10.1111/ijpo.12618 and other next-generation syntheses of similar trials to enhance power (see Seidler et al Guide to Prospective Meta-Analysis, BMJ 2019).

Thank you for this comment and suggestion. We agree that our assessment of redundancy is imperfect. We have added the following discussion of prospective meta-analyses to our discussion (pages 17-18, lines 312-317): “Prospective meta-analyses (PMA), which encourage harmonization of core outcomes and draw on individual participant data, can help clarify treatment effects and reduce research waste. In this way, individually underpowered studies can help address questions of significant clinical importance. Although successfully employed in other medical settings, PMAs were unfortunately not widely deployed in the early COVID-19 pandemic.” 

4. In view of 3, it is essential in the Discussion to acknowledge that (i) even individually underpowered trials can make a valuable contribution in addressing critically important questions regarding mortality if included in individual participant data meta-analyses and(ii) inability to assess how often this was happening is a major limitation of this study.

As above, in the discussion we have highlighted how (pages 17-18, lines 314-315): “individually underpowered studies can help address questions of significant clinical importance.” We have also added the following to the limitations section (page 19, lines 353-355): “We also did not assess the extent to which individual participant data were made publicly available (for example, through the Vivli platform), and subsequently incorporated into meta-analyses.”

---

## [Editor Report · Decision Letter 1]

20 Dec 2021

How Informative Were Early SARS-CoV-2 Treatment and Prevention Trials? A longitudinal cohort analysis of trials registered on clinicaltrials.gov

PONE-D-21-28808R1

Dear Dr. Waligora,

We’re pleased to inform you that your manuscript has been judged scientifically suitable for publication and will be formally accepted for publication once it meets all outstanding technical requirements.

Kind regards,

Dylan A Mordaunt, MB ChB, FRACP, FAIDH

Academic Editor

PLOS ONE

Additional Editor Comments (optional):

Thank you for your resubmission. This meets the criteria for publication.
---

## [Editor Report · Acceptance letter]

22 Dec 2021

PONE-D-21-28808R1 

How Informative Were Early SARS-CoV-2 Treatment and Prevention Trials? A longitudinal cohort analysis of trials registered on clinicaltrials.gov 

Dear Dr. Waligora:

I'm pleased to inform you that your manuscript has been deemed suitable for publication in PLOS ONE. Congratulations! Your manuscript is now with our production department. 

Kind regards, 

on behalf of

Dr. Dylan A Mordaunt 

Academic Editor

PLOS ONE